# CryoEM reveals the structure of an archaeal pilus involved in twitching motility

Matthew C. Gaines[1,2], Shamphavi Sivabalasarma[3,4], Michail N. Isupov [5], Risat Ul Haque[1,2], Mathew McLaren [1,2], Cyril Hanus [6], Vicki A. M. Gold [1,2], Sonja-Verena Albers [3,4,7] & Bertram Daum [1,2] ✉

Amongst the major types of archaeal filaments, several have been shown to closely resemble bacterial homologues of the Type IV pili (T4P). Within *Sulfolobales*, member species encode for three types of T4P, namely the archaellum, the UV-inducible pilus system (Ups) and the archaeal adhesive pilus (Aap). Whereas the archaellum functions primarily in swimming motility, and the Ups in UV-induced cell aggregation and DNA-exchange, the Aap plays an important role in adhesion and twitching motility. Here, we present a cryoEM structure of the Aap of the archaeal model organism *Sulfolobus acidocaldarius*. We identify the component subunit as AapB and find that while its structure follows the canonical T4P blueprint, it adopts three distinct conformations within the pilus. The tri-conformer Aap structure that we describe challenges our current understanding of pilus structure and sheds new light on the principles of twitching motility.

Archaea possess a large variety of filamentous cell surface extensions, which function in adhesion to (a)biotic surfaces, biofilm formation, DNA exchange, cell–cell recognition, exchange of nutrients and motility in liquid environments[1]. These include ABP filaments from *Pyrobaculum calidifontis*[2] and the *Sulfolobus acidocaldarius* threads, which structurally resemble bacterial type-I pili, but are likely assembled by a distinct mechanism[3]. Furthermore, *Methanothermobacter thermoautotrophicus* generates fimbriae, which are important for biofilm formation[4], and a structure of the filamentous archaeal DNA import machinery related to conjugative DNA transfer systems has also been reported recently[5]. Finally, archaea-specific filaments exist, such as unusual, barbed wire-like Hami of *Altarchaeum hamiconexum*[6] and canulae of *Pyrobaculum calidifontis*[2], which both appear to have important roles in adhesion to surfaces and other cells. However, perhaps the so-far best characterised are those homologous to T4P, which are common in archaea and bacteria.

Many archaeal genomes contain operons encoding T4P[7]. The complexes that assemble archaeal T4P are generally thought to be simpler than those found in bacteria, and mainly contain an ATPase, an inner membrane platform protein, and a set of pilins. The latter exhibit a canonical class III signal peptide that is processed via a PibD/ArlK peptidase protein prior to their assembly into the pilus fibre[8–11]. T4P are comprised of thousands of copies of the major pilin subunit(s), plus low-abundance minor pilins[12]. Major pilins make up the bulk of the filament, while minor pilins are thought to form cell-proximal or cell-distal cap structures and are often essential for the assembly of the pilus. Both the major and minor pilins adopt a tadpole-like structure consisting of an N-terminal α-helix (α1), followed by a globular β-strand-rich C-terminal head domain[13].

Archaella belong to the same superfamily as T4P and are arguably the best investigated archaeal filaments. The archaellum acts as a gyrating filamentous propeller that enables cells to swim through liquid media[14–16]. As is typical for T4P, the archaellum has a helical symmetry and is highly glycosylated[8,17–23]. Six structures of archaella have been solved by CryoEM to date[18,20–22,24]. These revealed that archaellins follow the structural blueprint that is similar to T4P,

[1]Living Systems Institute, University of Exeter, Exeter, UK. [2]Department of Biosciences, Faculty of Health and Life Sciences, Exeter, UK. [3]Institute of Biology, Molecular Biology of Archaea, University of Freiburg, Freiburg, Germany. [4]Spemann Graduate School of Biology and Medicine, University of Freiburg, Freiburg, Germany. [5]Henry Wellcome Building for Biocatalysis, Department of Biosciences, Faculty of Health and Life Sciences, University of Exeter, Exeter, UK. [6]Institute of Psychiatry and Neurosciences of Paris, Inserm UMR1266 - Université Paris Cité, Paris, France. [7]Signalling Research Centres BIOSS and CIBBS, Faculty of Biology, University of Freiburg, Freiburg, Germany. ✉e-mail: b.daum2@exeter.ac.uk

including the conserved hydrophobic α-helix and the more variable globular C-terminal domain. The N-terminal α-helices bundle to form the core of the filament, while the C-terminal globular domains face outside[18,20,21]. The archaellum can consist of one repeating subunit, or have complex heteropolymeric composition, as revealed by a recent structure from *Methanocaldococcus villosus*[22].

Non-rotary archaeal T4P are thought to function as adhesives to various biotic and abiotic surfaces, and to enable DNA exchange and intercellular communication[1,25]. Furthermore, archaeal T4P have been shown to serve as receptors for a range of archaeal viruses[26]. Hyperthermophilic strains such as *Sulfolobus islandicus* are infected by the rod-shaped viruses 2 and 8 (SIRV2 and SIRV8, respectively; family *Rudiviridae*)[27,28] and others, such as *Saccharolobus solfataricus* by the turreted icosahedral virus (STIV; family *Turriviridae*)[26,27]. Another well-studied pilus is the Ups, which is formed by various *Sulfolobus* species in response to UV-induced double strand DNA breaks and leads to species specific cell-cell aggregation[29,30]. This allows the cells to exchange chromosomal DNA for homologous recombination[31,32], putatively using a bacterial-like DNA-exchange apparatus.

In *S. acidocaldarius*, the Aap has previously been described as an adhesive filament that is important for biofilm formation, and a low resolution cryoEM map has been published[33]. Based on this map, it was for the first time suggested that, indeed, archaeal pili are homologous to bacterial T4P[33]. These pili are found in all *Sulfolobales* genomes and a comparison of known homologous adhesive pili from closely related organisms such as *S. islandicus* or *Saccharolobus solfataricus* shows that these filaments are highly glycosylated, possibly as a protective adaption to their harsh natural environments[34,35].

New interest in Aap was triggered through the findings of a recent study, showing that this pilus is involved in archaeal twitching motility[36]. Twitching motion is a coordinated process that begins with the extension of T4P from the cell surface. The pili then attach to a substrate, such as a solid surface or another cell, and retract, pulling the cell towards the attachment site. This process is repeated, resulting in a series of jerky movements that propel the cells across surfaces[37]. Twitching motility is particularly well studied in Gram-negative bacteria, including pathogenic and environmental species, and plays crucial roles in bacterial pathogenesis, biofilm formation, and nutrient acquisition[38-43]. In such species, this process is often driven by a complex molecular machine, which includes motor ATPases that provide the energy for pilus extension (PilB)[44] and retraction (PilT)[38,45], a cell-membrane integral platform protein (PilC)[45], an outer membrane-spanning conduit (PilQ)[46], and accessory proteins forming a periplasmic cage-like structure, called an "alignment complex", composed of PilM, N, O and P[45]. While archaeal genomes encode for homologues of the platform protein PilC and the assembly ATPase PilB, genes for retraction ATPases akin PilT, PilQ or the PilMNOP complex are not found in archaeal T4P assembly clusters[7]. Two pilins called *aapA* and *aapB* are present in the *S. acidocaldarius aap* pilus cluster. However, so far it was not clear whether both pilins form the pilus, as is the case for the archaellum of *M. villosus*[22], or if the filament is mainly composed of one of the two gene products.

Here, we determine the structure of the Aap, and show that it is solely composed of multiple copies of AapB. Strikingly, instead of using a second pilin, AapB adopts three distinct conformations within the pilus. This gives the Aap a unique, tri-conformer architecture which has previously not been seen in any bacterial or archaeal pilus. We hypothesise that this feature may have important implications for the mechanism of twitching motility.

## Results

### CryoEM and helical reconstruction of Aap filaments

Filaments of *S. acidocaldarius* strains lacking archaella and Ups (MW158) were sheared from cells at stationary phase and purified via CsCl gradient centrifugation. The resulting mix contained threads and Aap. The suspensions were plunge frozen on cryoEM grids, from which 6272 movies were recorded using a Titan Krios transmission electron microscope (TEM). CryoSPARC[47] was used for the entire image processing workflow. The raw micrographs revealed highly flexible Aap filaments with curvatures of up to 90 degrees (Supplementary Fig. 1). In contrast to the Aap, the archaella of the same organism show a slightly undulating superstructure and the threads are mostly straight (Supplementary Fig. 2).

Iterative 2D classification was used to separate the Aap from the thread filaments, the structure of which we solved previously[3]. Using the Helical Refinement function in CryoSPARC[47], we were able to reconstruct an initial unbiased 3.7 Å resolution map of the Aap without applying helical symmetry parameters (Supplementary Fig. 3 b). Aided by the building of an initial atomic model, this map was sufficient to deduce the helical parameters, which we identified as a helical twist of −39.9° and a rise of 15.4 Å. Applying these parameters in subsequent refinements finally resulted in a map with 3.2 Å resolution (Fig. 1, Supplementary Fig. 3 d, 4 c, d and 5). Local resolution estimates showed that the core of the map has a resolution of 2.8 Å, whereas at the periphery of the filament the resolution falls off to 3.8 Å, with an average global resolution of 3.2 Å (Supplementary Fig. 5). The map showed that the filament consists of multiple copies of tadpole-shaped monomers typical for T4P. In agreement with the resolution, large side chains could be clearly identified throughout.

Our structure thus revises the previously published ~ 9 Å resolution map of the *S. acidocaldarius* Aap[31], which was a result of the application of 136.9° twist and 5.7 Å rise. Moreover, the helical parameters identified by us differ from those published for homologous of adhesive T4P of *S. islandicus*[34] and *S. solfataricus*[35], with published values of around 105° twist and 5 Å rise. Performing helical symmetry search in cryoSPARC resulted in a match with apparent values of 106.7° twist and 5.12 Å rise. However, when applying these parameters, the resulting map was of low quality (Supplementary Fig. 3c) and contained poorly defined core α-helices that could not be modelled. The layer line profile of this 5.12 Å rise map was altered in comparison to that of the non-biased map, particularly when viewed along the Z axis (Supplementary Fig. 3b, c). In contrast, applying −39.9° ($106.7 \times 3 = 320.1 = 360 - 39.9 = -39.9°$) twist and 15.4 Å rise produced layer line profiles that closely resembled the non-biased version, with far superior detail and improved resolution (Fig. 1; Supplementary Fig. 3d). Further evaluation of our unbiased map showed that this filament consists of three stacked left-handed helices in distinct conformations (Fig. 2a).

### Composition and structure of the Aap

In *S. acidocaldarius*, the *aap* gene cluster encodes for two pilins, AapA and AapB (Supplementary 6a). Previous experiments indicated that neither the deletion of *aapA* nor *aapB* led to loss of Aap filament formation[33]. However, these findings were based on low-resolution TEM data that precluded the clear distinction between Aap and archaella. Re-examining Δ*aapA*, Δ*aapB* and Δ*aapAB* mutants with more advanced TEM equipment and improved imaging conditions, we found that the Δ*aapA* strain still assembles Aap (Supplementary Fig. 7a), whereas Aap were lost in the Δ*aapB* knockout (Supplementary Fig. 7b), as well as the double mutant Δ*aapAB* (Supplementary Fig. 7c).

We therefore wondered if the filament could be composed of a mixed population of AapA and AapB, similar to the recently published structure of the archaellum of *Methanocaldococcus villous*[22]. By building the atomic model of the Aap pilus ab initio aided by large amino acid side chains, as well as glycosylation sites that differ in AapA and AapB (Supplementary Fig. 8), we found that only the sequence of AapB could be reconciled with our map. This suggests that the filament is entirely comprised of AapB (Fig. 2a, b). We corroborated our findings by predicting the structures of AapA and AapB using Alphafold[48]. The prediction of AapB closely matched our ab initio model. In contrast,

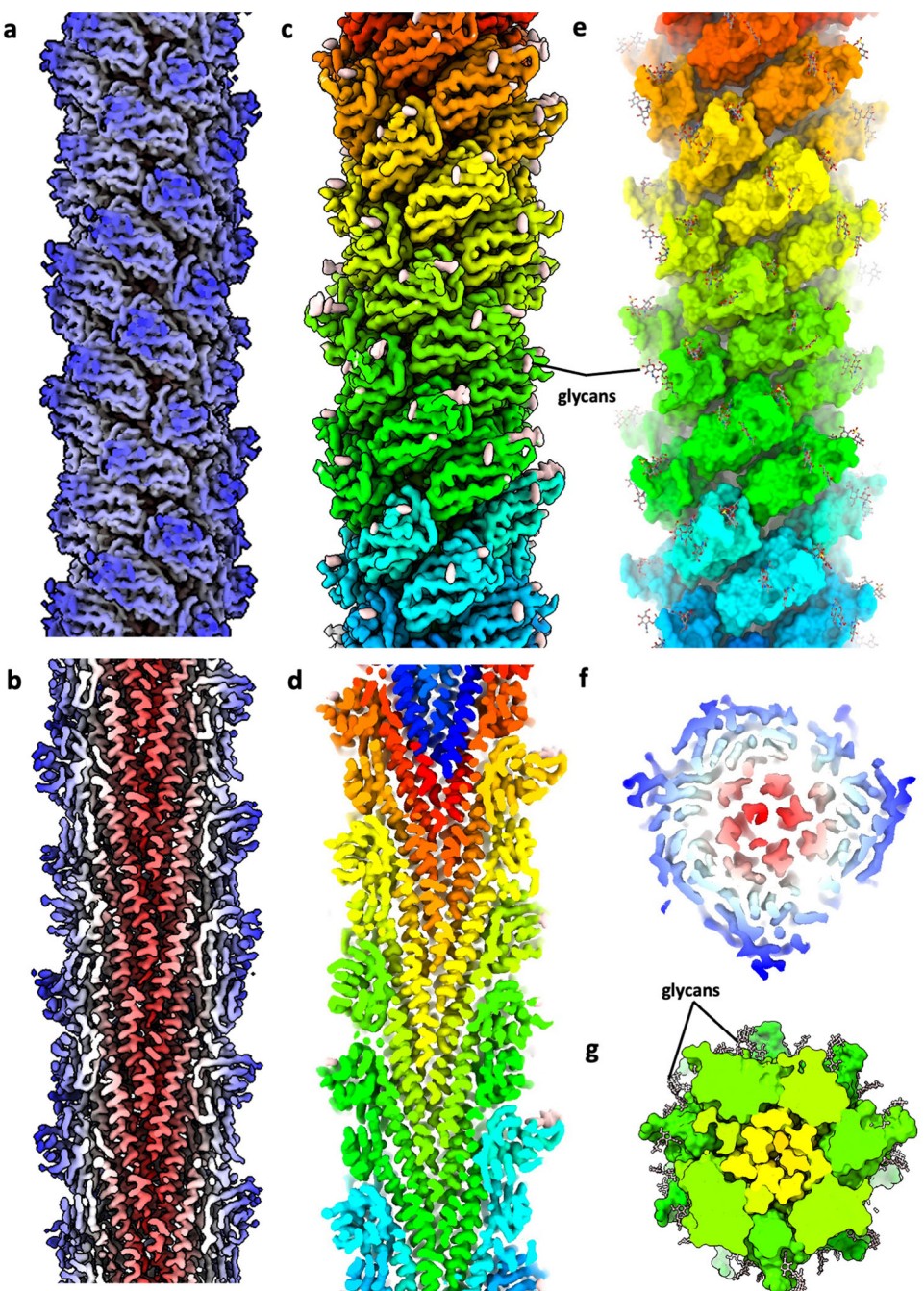

**Fig. 1 | Helical reconstruction and atomic model of the *S. acidocaldarius* Aap.** Surface view **a**, **c** and cross-section **b**, **d** of the cryoEM map of the Aap. In **a**, **b**, the maps are coloured by cylinder radius (α-helical core, red; β-strand rich globular domains, in shades of blue). In **c**, **d**, the map is coloured by subunit. Glycans are shown in peach. **e** atomic model with protein in surface view and glycans in ball and stick representation. **f**, end-on view of the map coloured by radius (α-helical core, red; β-strand rich globular domains, in shades of blue) and **g**, end-on view of the model in surface representation (subunits in shades of green and glycans in stick representation). Scale bar, 60 Å.

the predicted structure of AapA did not fully comply with our ab initio model (Supplementary Fig. 8) or our map. In addition, our assignment of AapB was confirmed by ModelAngelo[49]. Based on our map, the software suggested a protein sequence with high sequence similarity to AapB, thus confirming AapB as the sole subunit composing the filament (Supplementary Fig. 9). In line with this, transcriptome analysis revealed that *aapA* is expressed in lower amounts than *aapB*[50] (Supplementary Fig. 6b). It is thus conceivable that this protein acts as a minor pilin, which may either assemble into a cap structure for the pilus or could reside within the cellular membrane and function as a

part of the Aap assembly machinery. Interestingly, Charles-Orszag et al.[51], demonstrated that AapA-deficient mutants show significantly reduced twitching motility.

Each AapB subunit consists of an N-terminal α-helix tail, followed by a globular (head) domain. The head domain contains 8 β-strands that fold into two β-sheets (Supplementary Fig. 10). T4P are N-terminally processed by a class-III signal peptidase (PibD/ArlK in archaea)[52]. In accordance with this, we found that 15 N-terminal amino acids are missing in the mature protein. Indeed, the FlaFind server[53] predicted a peptidase processing site between A15 and L16.

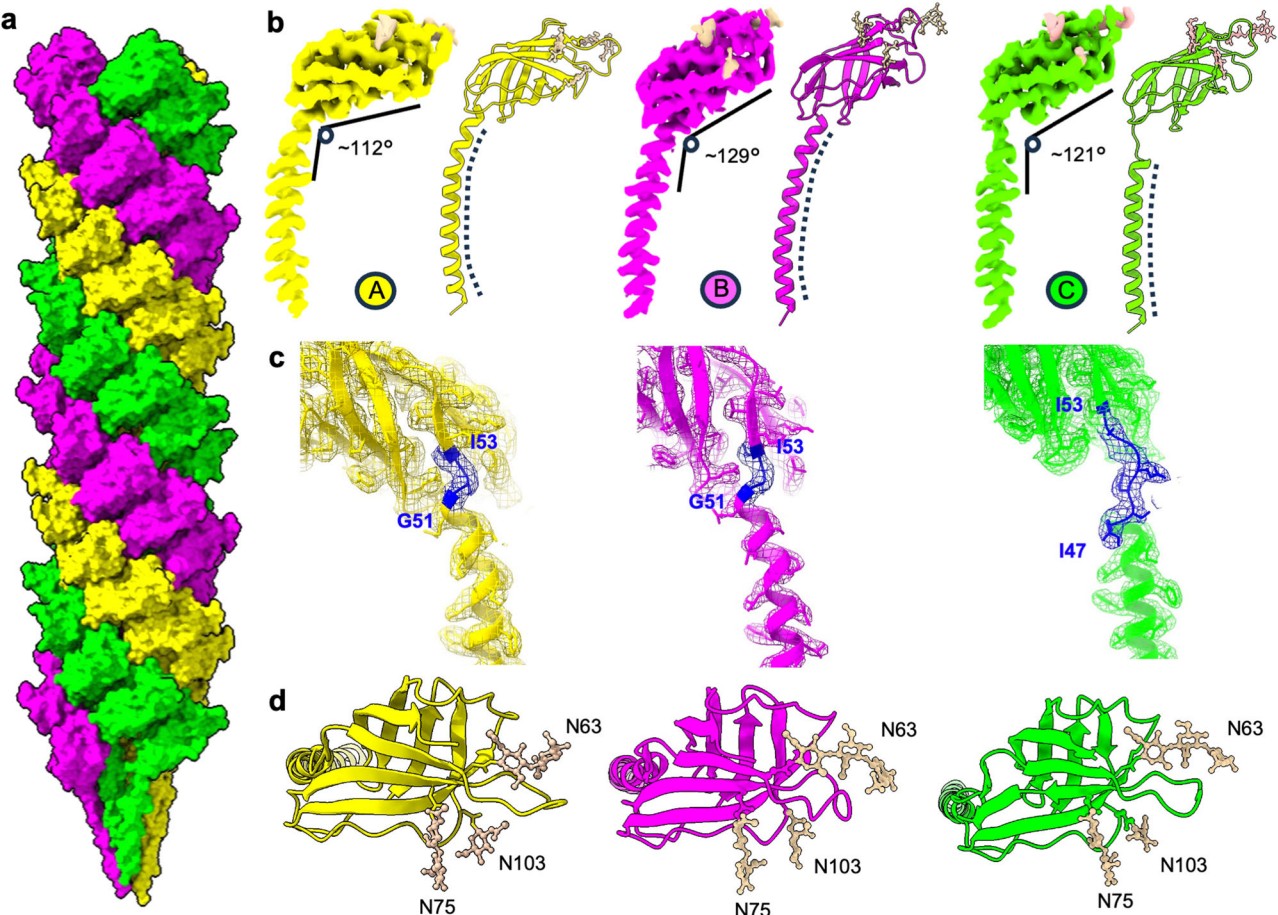

**Fig. 2 | AapB adopts three conformations within the Aap. a** structure of the *S. acidocaldarius* Aap in surface representation. The pilus is a 3-start helix and consists of three left-handed helical strands (magenta, yellow and green). Each of the three component helices is composed of multiple copies of AapB which adopt one of three conformations (A, yellow; B, magenta and C, green). **b** AapB subunits in their three conformations (A, B and C) shown as maps (left) and models (right). The three conformations differ in the angle between the head and α-helix tail close to the head, which measure ~112°, ~129°, and ~121° for A, B and C, respectively, as well as the curvature of the helix (dotted line). **c** Closeups of atomic models (ribbons) fitted into the cryoEM map (mesh). The hinge regions between heads and tails are shown blue. A and B have a similar, short hinge between G51 and I53, where the α-helix transitions into the first β-strand of the head (β1). In conformation C, the α-helix is "melted" into a long loop region, extending the hinge from I47 to I53. **d** Closeups of the head domains with glycans linked to N63, N75 and N103 shown in stick & ball representation (peach).

Careful analysis of the filament's structure revealed that the AapB monomers exist in three distinct conformations (henceforth designated as conformations A, B and C; Fig. 2a, b) These differ in the angle between the head and the tail domain with values of ~112°, 129°, and 121° for conformations A, B and C, respectively (Fig. 2b; Movie 1). The RMSD between conformations A and B is the highest with a value of 2.41 Å, whereas the RMSD between conformations A and C, and B and C are 1.94 Å and 1.60 Å, respectively (Supplementary Fig. 11). Interestingly, the conformation of AapB depends on its location in the filament. Each distinct conformation repeats along one of the three left-handed 3-start helices of the pilus, meaning that each of these 3-start helices harbours exclusively one of the three conformations (Fig. 2a). This leads to a structurally unique fibre that consists of three stacked helices with distinct conformations.

Closer inspection of the three conformations revealed the distinct intra-subunit angles are due to differences in the hinge region of AapB. This hinge is a hallmark of all T4P and is characterised by a loop of one to several amino acids in length that joins the N-terminal α-helix to the first β strand (β1) of the head domain[54] (Fig. 2 c). In conformation A and B this hinge is short and lies between G51, the final amino acid of the α helix and I53, the first amino acid of β1. This hinge is slightly more open in conformation B, thus causing the wider angle between the head and the tail domains. In conformation C, the hinge region spans from I53 to

I47, across a stretch of four additional amino acids where the α-helix is "melted" into a loop. As a result, the angle between head and tail is 121°.

Moreover, the distinct hinge of conformation C causes the tail domain to stretch out, meaning that it reaches further into the centre of the filament (Fig. 3). This results in a unique organisation of the filament's core. Whereas the N-terminal residues (L16) of conformation C form a central axis, the same residues of conformations A and B spiral around this central axis in a left-handed fashion (Fig. 3 a–h). This contrasts with the organisation of the core of archaella, where all N-terminal residues are equidistant to the centre of the filament (Fig. 3 i, j).

Conformations A, B and C are organised into structurally unique left-handed 3-start helices Figs. 2a and 4a). When the Aap is viewed in the right-handed 7-start or 4-start directions, the conformations alternate (A,B,C,A,…)—an architecture that is not seen in mono-conformer structures, such as that previously reported for *S. islandicus* LAL14 pilus[34] (Fig. 4b).

The archaellum of *M. villosus* is a heteropolymer[22], it consists of two alternating subunits (ArlB1 and ArlB2). In this structure, no homopolymeric strands containing only one subunit occur in any helical direction. While the left-handed 3-start helices follow the pattern ArlB1, ArlB2, ArlB1 and so forth, different repeats are seen in right-handed 7- or 10-start component directions (Fig. 4c).

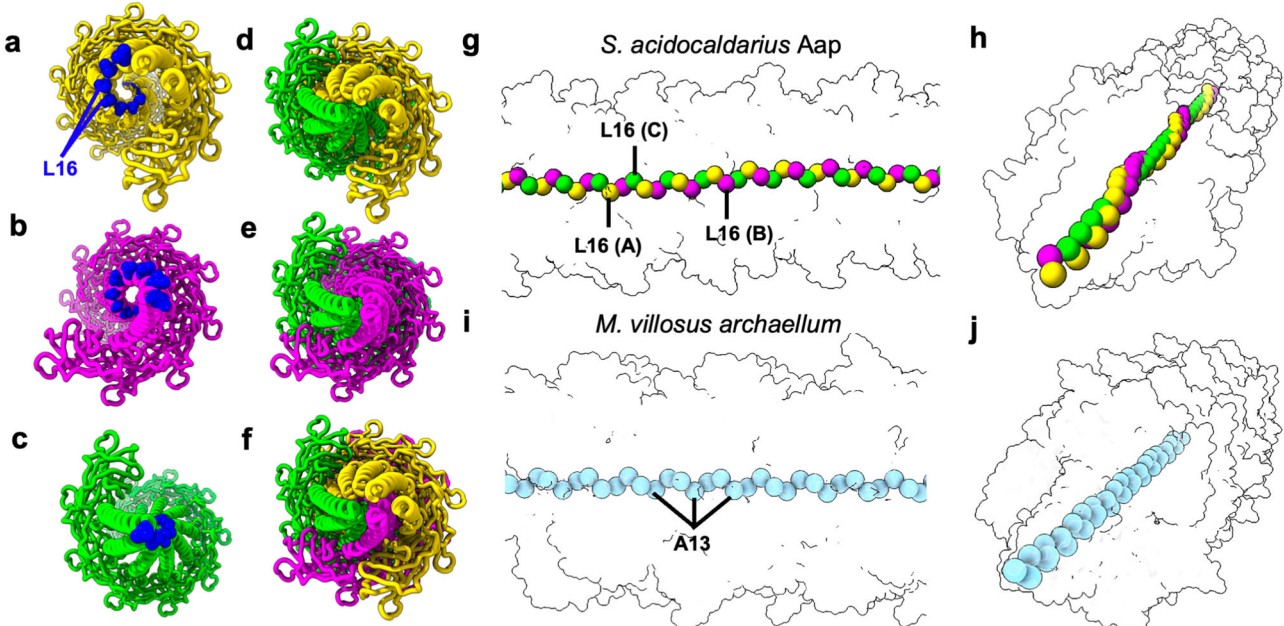

**Fig. 3 | The conformational heterogeneity of the Aap impacts the packing of its core. a–c** separate end-on views of the conformations A (**a**, yellow), B (**b**, magenta) and C (**c**, green), as they assemble in the filament. The N-terminal residues (L16) are shown as blue spheres, the rest of the protein is shown in liquorice representation. While the N-terminal (L16) residues of conformation C form a tight spiral around the central axis of the filament (c), they are more distant from the centre in conformations A and B **a**, **b**. **d–f** end-on views of the conformation combinations of A and C (**d**), B and C (**e**) and A, B and C (**f**). **g**, **h** Packing of N-terminal Cα atoms (L16 residues) for conformations A (yellow), B (magenta) and C (green) in side **g** and diagonal **h** views. The L16 residues of conformation C (green) line up along a central axis. The L16 residues of conformations A (yellow) and B (magenta) spiral around this axis. **i**, **j** the organisation of N-terminal Ala13 residues in the *M. villosus* archaellum[23], shown as light blue spheres (**i** side view; **j** diagonal view). In contrast to the Aap, all N-termini are equidistant to the central axis.

Interestingly, the α-helical tails within the core of the *S. acidocaldarius* Aap are significantly curved, with the concave side of each α-helix facing the filament's periphery (Fig. 4d–f). This leads to a screw-like organisation of the filament's core (Fig. 4d, e). As Alphafold predictions for a single AapB subunit suggest a straight α-helix (Supplementary Fig. 9), it is possible that the N-termini of AapB become distorted upon integration into the pilus. A similarly (albeit to a lesser extent) distorted organisation is seen in the structure of the *S. islandicus* LAL14 pilus[34] (Fig. 4g–i). Very recently the structure of another Aap, the REY15A filament of *S. islandicus*[55], has been solved. The REY15A Aap has similar helical parameters as the *S. acidocaldarius* Aap. It also consists of a one subunit in three conformations and the same screw-like core superstructure as in the *S. acidocaldarius* Aap is observed.

In archaella on the other hand, the subunits are slightly bent the other way, with the convex sides of the α-helices facing towards the filament's periphery (Fig. 4j–l)[18,20–22]. Thus, the core of the archaellum does not have a screw-like superstructure (Fig. 4j, k). Even though the two subunits of *M. villosus* (ArlB1 and ArlB2) have a distinct sequence and glycosylation pattern[22], they do not differ in the angle or the hinge region between head and tail (Fig. 4i). Therefore, the distorted, tri-conformer organisation appears to be a unique feature of Aap, at least in *S. acidocaldarius* Aap, as well as the recently reported REY15A pilus of *S. islandicus*[55].

### The Aap is posttranslationally modified with highly flexible N-glycans

During atomic model building, we identified three glycans per AapB subunit (Fig. 2b, d). In the cryoEM map, these became apparent as dead-end protrusions which had characteristic features of archaeal glycans (Supplementary Fig. 12a–c)[3,20,22]. The glycan densities originated from asparagine residues, (N48, N60, and N88), which were all part of a consensus N-glycosylation sequon (NXS/T; Supplementary Fig. 12a–c). For N-linked glycosylation, *S. acidocaldarius* utilises tri-branched hexasaccharides, featuring the rare 6-sulfoquinovose sugar. In contrast, no evidence of O-glycosylated Serine or Threonine residues was found.

We previously gained structural insight into this glycan by building the sequence known from mass spectrometry[56–58] into particularly well-defined glycan densities seen in the *S. acidocaldarius* thread[3] (Supplementary Fig. 12e–i). As *S. acidocaldarius* only creates one type of glycan[56], we built the same glycan sequence into the corresponding densities of the Aap. However, as the glycan densities of the Aap were less well defined than in the thread, we only modelled the sugar units for which a density was clearly resolved. Notably, we found a fourth Asparagine (N114), which resides within a N-glycosylation consensus sequon, but did not show any glycan density (Supplementary Fig. 12d). This differs from our structures of the *S. acidocaldarius*, thread and *M. villosus* archaellum, where usually every surface-exposed N-glycosylation sequon is glycosylated[3,22]. The absence of the glycan at position N114 may be explained by potential inaccessibility of the site in the preprotein, or previous findings, suggesting that AglB, the enzyme that glycosylates proteins in *Sulfolobus*, is promiscuous[59,60].

When comparing the glycan densities of the Aap pilus with those of the thread filament, it becomes clear how much more well-defined the glycans are in the threads (Supplementary Fig. 12e–i)[3], even though both filaments were resolved at similar global resolution and using the same sample. While all six sugar units of the hexasaccharide were resolved in the threads (Supplementary Fig. 12e–i), the map of the Aap contained only densities for the first one or two sugar residues (Supplementary Fig. 12a–e). Densities for the two N-acetylglucosamine (GlcNAc) residues, the connecting mannose (Man), glucose and 6-sulfoquinovose molecules were not resolved. Similar observations were made in the published cryoEM maps of *P. arsenaticum, S. solfataricus*[35], and *S. islandicus*[34] T4P. To determine whether this difference was due to various degrees in glycan flexibility, we performed

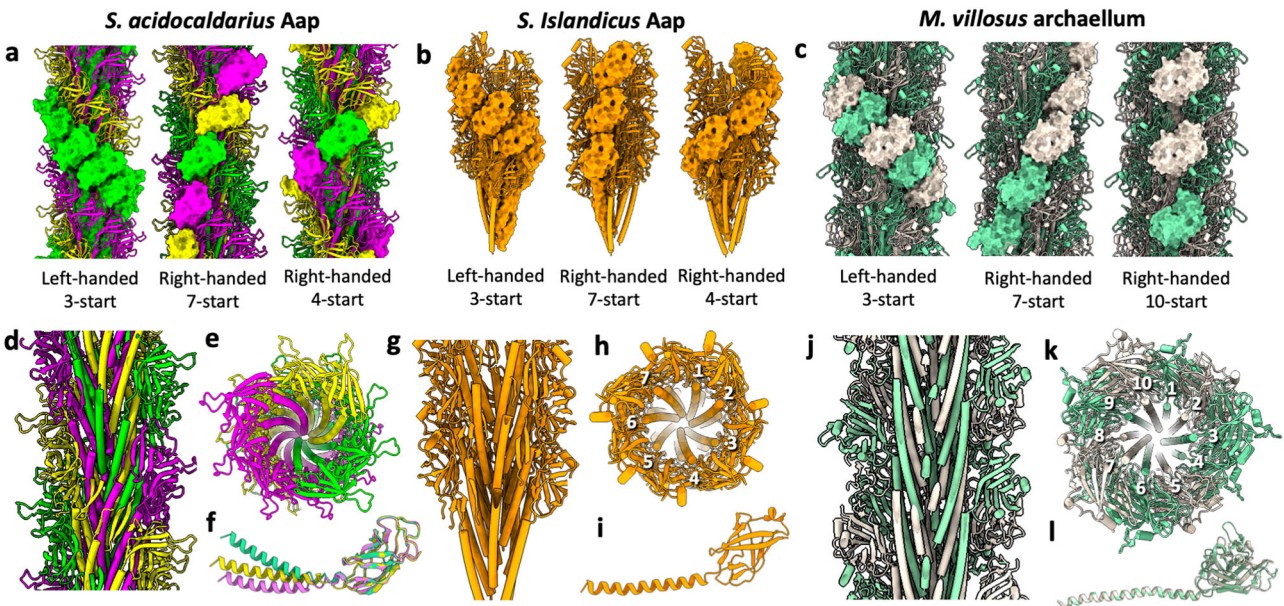

**Fig. 4 | The *S. acidocaldarius* Aap in comparison with pili and archaella. a** The tri-conformer structure of the *S. acidocaldarius* Aap with one left-handed 3-start helix (left); one right-handed 7-start helix (middle), and one right-handed 4-start helix shown in solid representation. Each left-handed 3-start helices contain one AapB conformation, whereas the right-handed 7-start and 4-start helices consist of a repeating sequence of conformations A,B,C,A,…. **b** Single conformer structure of the *S. islandicus* LAL14 AAP filament (PDB 6NAV[35]) with one left-handed 3-start helix (left); one right-handed 7-start helix (middle), and one right-handed 4-start helix (right) shown in solid representation. The filament's subunits are the same in any helical direction. **c** Two-subunit structure of the *M. villosus* archaellum (PDB 7OFQ[23]). ArlB1 is shown in white and ArlB2 in green. Within the archaellum, left-handed 3-start helices (left) consist of alternating subunits with the sequence ArlB1, ArlB2, ArlB1,… Right-handed 7-start helices (middle), and right-handed 10-start helices (right) show different heteropolymeric repeat patterns. None of the component helices in the *M. villosus* archaellum are homopolymers. **d** Longitudinal cross section and **e** end-on view of the Aap showing that bent α-helices in the filament's core generate a screw-like superstructure. **f** The three AapB conformers superimposed. The α-helices of the three conformers show different degrees of curvature. The convex sides of the α-helices face the core of the filament. **g** Longitudinal cross section and **h** end-on view of the *S. islandicus* (LAL14) Aap. The filament's core shows a less pronounced screw superstructure. **i** The subunit of the *S. islandicus* (LAL14) Aap appears to be less curved than any of the AapB conformers. **j** Longitudinal cross section and **k** end-on view of *M. villosus* archaellum. The core of the filament does not show a screw superstructure. **l** The two subunits ArlB1 (white) and ArlB2 (green) superimposed. Both subunits show the same slight curvature. In contrast to the Aap, the convex faces of the α-helices are oriented towards the outer sheath of the archaellum. While cores of the *S. islandicus* (LAL14) Aap (**h**) and the *M. villosus* archaellum (**k**) show clear 7-stranded, or 10-stranded organisation, respectively, this does not appear to the case for *S. acidocaldarius* Aap **e**, likely due to its 3-conformer structure.

Glyocoshield analysis[61], a molecular dynamics simulations approach focussed on modelling all possible glycan conformations on the surfaces of the Aaps and threads (Fig. 5a–d). This analysis revealed that the glycans in the thread indeed adopt fewer conformations than in the Aap pilus. This is particularly true for the thread glycan bound to Asn146, which is wedged inside a cleft between two adjacent thread subunits. Because of this reduced mobility of thread glycans, their structure could be reconstructed in helical processing, while the highly flexible Aap glycans largely averaged out.

**Filament flexibility**

It has previously been suggested that the flexibility of glycans correlates with the flexibility of proteins[61], suggesting that Aap should be more flexible than the thread filaments of the same organism[3]. Indeed, our cryoEM micrographs (Supplementary Figs. 1 and 2), as well as 2D classes (Fig. 5e, f) indicate that threads are typically straight, while Aap are able to bend considerably. To understand the structural basis of the apparent differences in filament stiffness, we performed 3D Variability Analysis (3DV) in CryoSPARC and built atomic models for the generated output (Supplementary movie 2).

While this 3DV approach could only probe a limited conformational range, the analysis confirmed our 2D classification data. Aap appear more flexible compared to the threads (Supplementary movie 2). This is likely functionally important, as threads are mainly thought to facilitate cell-cell adhesion and biofilm formation[3], while Aap are involved in twitching motility[36].

The observed differences in flexibility can be explained with the distinct intermolecular interactions that hold Aap and threads together. In the Aap, the main intermolecular contacts are found in the form of hydrogen bonds between the C-terminal head domains and hydrophobic interactions between the N-terminal tails. As in the archaellum, the Aap heads are linked to the tails by a conserved hinge region and are thus free to move with respect to the tails[22]. In the Aap, this hinge region varies depending on the conformational state of AapB (Fig. 2c). The three distinct hinge structures may lead to different stiffnesses of the 3 component helices. In addition, the hydrophobic interactions in the core of T4P and archaella act as molecular grease that enables the tails of the subunits to slide past each other[22], and this can be seen in the 3DV analysis of the Aap (Supplementary movie 2). The threads, on the other hand, do not have a hydrophobic core that could provide the same degree of flexibility. Instead, the subunits are interlinked by donor strand complementation (DSC), where the N-terminal tail of one subunit integrates into and completes the β-sheet of the next subunit (n+1) along the chain[3]. β-sheets are rich in hydrogen bonds and thus do not allow for significant flexibility. Furthermore, there are no pronounced intramolecular hinges in the thread and an isopeptide bond is formed between subunit n and n+2[3], covalently interlinking the subunits. Finally, the glycan associated with N146, which is wedged into a cleft between two adjacent subunits (Fig. 5d, Supplementary Fig. 12i), likely restricts the movement of the thread subunits with respect to each other. No sterically wedged glycans occur in the Aap, where the glycans are free to explore a large

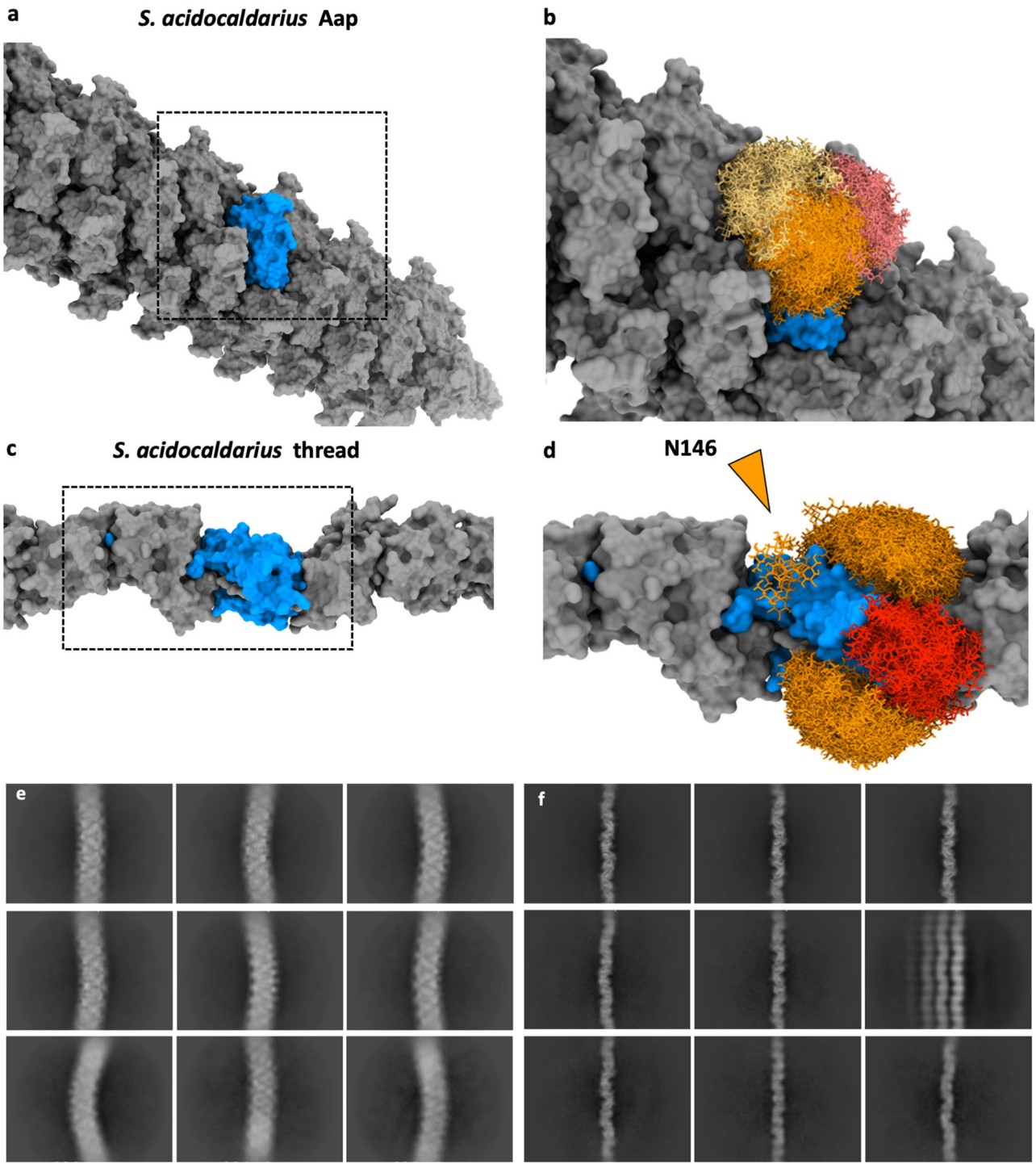

**Fig. 5 | Glycosylation and flexibility of Aap compared to threads of *S. acidocaldarius*. a, b** Glycoshield simulation[62], showing the large conformational space that the surface glycans (yellow, orange, red) occupy in a singular AapB subunit (**a** without and **b** with glycans for comparison). **c, d** Glycoshield simulation for the four glycans in a singular *S. acidocaldarius* thread subunit (**c** without and **d** with glycans; PDB 7PNB[3]). In contrast to the Aap, one of the thread glycans (N146, orange arrowhead) occupies a limited conformational space, as it is wedged between the interface of two subunits. **e, f** 2D classes of Aap (**e**) and threads (**f**) show differences in stiffness between both filaments.

range of conformations, thus allowing for a highly flexible Aap filament.

### Aap are conserved among some Crenarchaea

To investigate the conservation of Aap, we used SyntTax[62] and ConSurf [63] to search for AapB homologues in related archaeal Thermoprotei species and found six conserved proteins (Supplementary

Fig. 13). Henche et al.[33]. previously suggested that two of these species (*S. islandicus* and *S. solfataricus*) would not assemble Aap, as their AapB homologues are not located near the machinery genes. However, in recent years, the structures for both these Aap filaments were solved[34,35]. The helical parameters for both pili were determined at 5 Å rise and -105° twist, suggesting single conformer filaments, in contrast to our tri-conformer structure with 15 Å rise and −39.9°

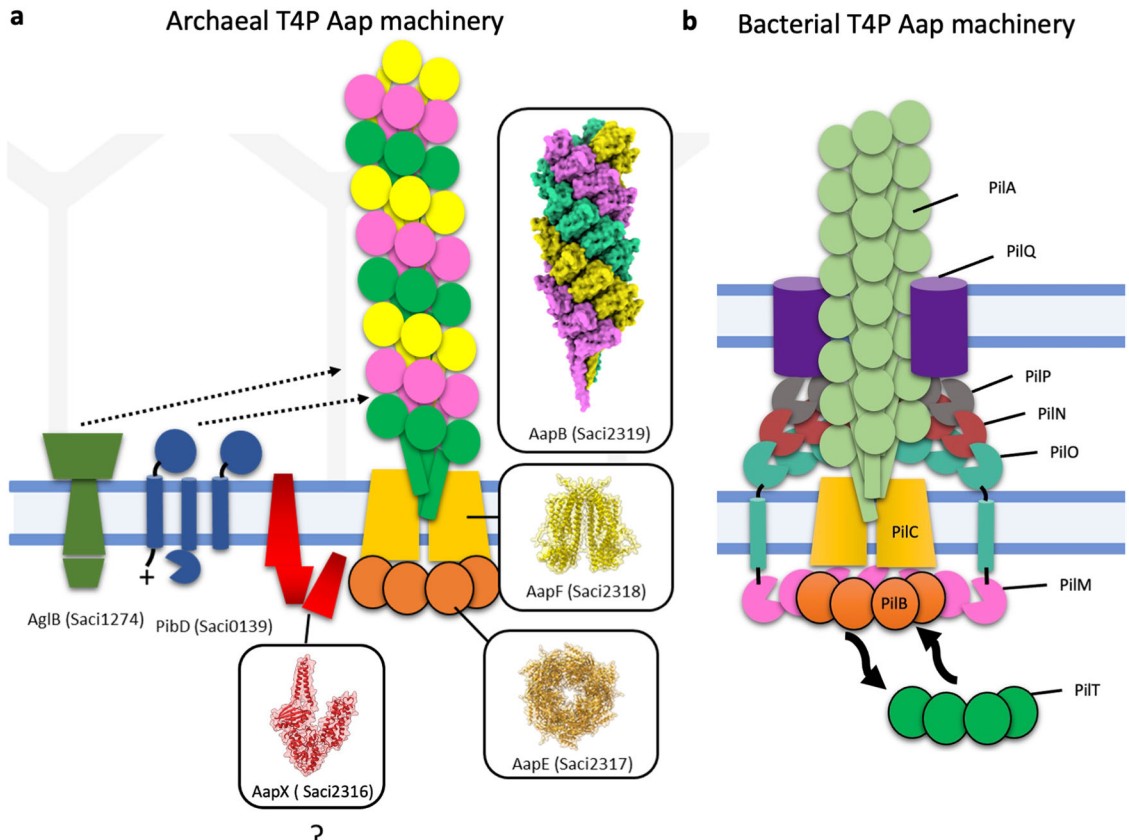

**Fig. 6 | Model comparing the archaeal and bacterial twitching apparatus.**
**a** Model of the Aap machinery from *S. acidocaldarius*. The pilus is composed of AapB and assembled by the putatively dimeric, and membrane-integral platform protein AapF. The latter binds to the cytoplasmic ATPase AapE. AapX is also likely part of the machinery, but its function is unknown. The signal peptidase PibD is responsible for the cleavage of the AapB pre-pilins, which primes them for pilus integration. AglB is the enzyme that glycosylates cell-external proteins, including AapB. AapB pilins assemble into the tri-conformer pilus. If the pilins adopt their three conformations prior to or during pilus assembly is unknown. **b** Model of the bacterial T4P machinery from *Neisseria gonorrhoeae*[94]. The PilA filament is assembled by the membrane-integral platform protein PilC and transverses through a PilQ pore, which is integrated in the outer membrane and peptidoglycan layer. In the cell wall, rings of accessory proteins (PilO, N and P) surround the filament. PilO extends down through the cell membrane to bind a ring of PilM, which encompasses the pilus assembly ATPase PilB. The latter is substituted for the disassembly ATPase PilT to rapidly retract the pilus and thus drive twitching motility. So far, only single conformer pili are known in bacteria.

twist. Despite this, all identified homologues are highly conserved on the amino acid level, and so we wondered if structural differences were evident in comparison with our tri-conformer Aap filament from *S. acidocaldarius*.

Multisequence alignments revealed almost identical N-terminal α-helices in all identified homologues, as well as several key glycine residues throughout the entire sequence. In all cases, Alphafold2[48] suggested almost identical structures when compared to *S. acidocaldarius* AapB (Supplementary Fig. 13a, b). Apart from the N-terminal tails, regions of high similarity included the core-facing β-strands of the head domain, with greater variability in the solvent-exposed strands and the loops connecting them. The nearly interchangeable α-helices indicate that these homologues may in fact all follow the helical parameters identified for the *S. acidocaldarius* Aap. Thus, the homologues may in fact also form pili that adopt similar tri-conformer structures, as it is the pilin tails that dictate this feature. In contrast, nuances such as the glycosylation sites, which are highly variable (Supplementary Fig. 13d), suggest that the position (and likely sequence) of the glycans play a key role in adapting each species to its environment or function. The widespread nature of the Aap raises the question of whether twitching motility is equally common in archaea.

### A model of the Aap Machinery

*AapB* is encoded in a gene cluster, together with putative machinery components *aapA, aapX, aapE and aapF* (Supplementary Fig. 6a). We used Alphafold2[48] to predict the structures of these proteins and, together with our solved structure of the Aap, integrated them into the current working model of the Aap machinery (Fig. 6a), which appears far simpler than the bacterial T4P machinery (Fig. 6b).

AapF is predicted to be a transmembrane protein, functioning as a platform during filament assembly, anchoring the Aap to the membrane, and connecting it to the ATPase AapE, which likely provides the energy for pilus assembly. AapF has previously been shown to be partially homologous to the membrane integral T4P platform protein PilC from *T. thermophilus*[64], (Supplementary Fig. 14a), as well as the archaellum assembly platform ArlJ from *S. acidocaldarius*[65], (Supplementary Fig. 14b). ArlJ is predicted to be a dimeric protein[65], and so we modelled AapF in the same manner. Deep TMHMM[66] predicted that AapF is a 9-transmembrane helix protein, with a bulky N-terminal domain in the cytoplasm and short periplasmic loops (Supplementary Fig. 15). Comparing AapF with ArlJ showed the greatest variability in the N-terminal region, suggesting a potential role in determining the proteins' substrate selectivity (pilin for ArlF vs. archaellin for ArlJ) (Supplementary Fig. 14e).

AapE is an ATPase predicted to function in Aap assembly, twitching motility, and potentially disassembly of the Aap filament[1,33]. Localised in the cytosol, AapE interacts with the platform protein AapF. AapE shows sequence similarity with the archaellum assembly ATPase ArlI (Supplementary Fig. 16a)[67]. This is reflected by the Alphafold2 prediction for AapE, which is consistent with the structure of the AAA

ATPase (Supplementary Fig. 16b). Indeed, it has previously been shown that the homologue from *S. solfataricus* functions as an ATPase[67]. As is common for AAA ATPases, ArlI is a hexameric protein[68], and so we modelled AapE as a hexamer (Supplementary Fig. 16c, d). Homologues of AapE and AapF were found in all six archaeal strains that also encode for an Aap-like pilin. Both genes always appear next to each other in the genome, although not necessarily near to the subunit homologue of AapB[33].

Alphafold2[48] predictions and PDBe-fold analysis of AapX indicated a membrane-anchored protein (Supplementary Fig. 17). Homologues of the *aapX* gene are also found in all other organisms that contain an *aapB* homologue and are located approximately 10 genes upstream from the putative ATPase and platform protein counterparts[33]. Interestingly, AapX has a transmembrane helix, which is absent in the other species investigated. Instead, these homologues have a FAD binding domain (Supplementary Fig. 17c, d), which is not present in *S. acidocaldarius*. As with *aapE* and *aapF* mutants, knockouts of *aapX* do not express Aap filaments[33]. Thus, while the function of AapX is unknown, it is necessary for the Aap assembly and likely a part of the Aap machinery.

## Discussion

Here we present the structure of the Aap pilus from *S. acidocaldarius* at 3.2 Å resolution. We find that the subunits of each left-handed 3-start helix exist in a distinct conformation. Conformational variability on the subunit level has recently been observed in flagella and archaella[24], as well as the bacterial flagellar hook of *Salmonella enterica*[69]. The FlgE hook of *S. enterica* consists of 11 subunits per turn, with each monomer inheriting a subtly different conformation that leads to the observed bending of the hook[70]. Similarly, it has been suggested that in archaellar and flagellar filaments from various species, conformational differences in the subunits generate the supercoiling necessary for swimming propulsion[22,71]. Whilst these observations explain how filaments adopt minimum energy states in clockwise vs anticlockwise rotation, the conformations found here in the Aap pilus are entirely different. In the Aap, we observe a 3-start helix configuration, where each of the 3 component helices adopts an independent structure that is likely consistent through the entire length of filament. The three conformations are largely established through different structures of the hinge regions between the N-terminal tails and the C-terminal heads of the AapB subunits.

In bacteria, pilus retraction is thought to be triggered as the pilus touches a surface. Presumably, a retrograde signal then ripples from the distal adhesion point to the pilus' assembly machinery in the cell envelope[72,73]. This has been proposed to activate pilus retraction, whereby the filament disassembles into its formerly constituent subunits, which concomitantly transfer from the disassembling filament into the cell membrane[72–74]. For efficient twitching, inter-subunit interactions within pili must be sufficiently strong to enable the assembly of a pilus, but also sufficiently weak, to allow swift disassembly during retraction. Thus, pili that are involved in twitching must exist in a metastable state.

Indeed, point mutations that weaken the subunit interactions between the major pilins of *Vibrio cholerae* decrease the stability of the filament, facilitate pilus disassembly and thus lead to faster retraction compared to the wild type, even without the aid of any ATPase activity[75]. The unusual triple conformer helix of *S. acidocaldarius* Aap may have evolved to aid pilus retraction. Notably, the *S. acidocaldarius* Aap shows an unusually twisted core structure. Herein, the α-helices of AapB are significantly bent, with each AapB conformer showing a unique curvature. This contrasts with the straight α-helix that is suggested for AapB by Alphafold2 (Supplementary Fig. 9), or that is seen in archaellins.

As the lowest energy of an α-helix is when it lies in a straight conformation, the monomers within the Aap appear to exist in an elevated energy state, perhaps comparable to a loaded spring. It is conceivable that this energy could be spontaneously released (for example as the pilus touches a surface), thus eliciting the elusive retrograde signal through the pilus, or the pilus' collapse into the cell membrane. The latter could exert a force capable of pulling the cell forward. In accordance with this hypothesis, in both *P. aeruginosa* and *N. gonorrhoeae*[76] AFM-induced tension resulted in conformational changes in the filaments. The distinctive structure of the Aap also necessitates heterogeneous interfaces between the three different pilin conformations. Applying the helical parameters that were previously published for similar pili from *S. solfataricus* and *S. islandicus*[34,35], results in a pseudo-homogeneous structure with of a straight core. Such a hypothetical organisation would lead to a closer packing between the central α-helices of AapB than seen in the heterogenous *S. acidocaldarius* Aap, and thus could render pilus disassembly less efficient due to stronger interactions.

Delving into the evolution of *S. acidocaldarius* Aap from the last universal common ancestor, the closest related group of bacteria to archaeal species are those that produce Tad pili[16]. Notably, the Tad pili, along with gram-positive competence pili, Type II secretion filaments and some Type IVb filaments all possess only one ATPase, yet are all capable of filament retraction[77]. The Tad pilus ATPase in *C. crescentus*, CpaF, has been shown to power the extension and retraction of tad pili —a mechanism that was also found in other species[77]. This finding refuted the previously hypothesised retraction mechanism centred around the minor pilins encoded in the pilus operon, showing that the minor pilins may be dispensable for pilin formation within retraction-deficient backgrounds[13]. Likewise, only one type of ATPase can be found for the Aap system of *S. acidocaldarius*, as well as related archaeal species, and only a single ATPase is currently known to drive the assembly and rotation of the archaellum, both in clockwise and anticlockwise rotation. It therefore appears plausible that Aap assembly and retraction could also occur through the bifunctional ATPase in *S. acidocaldarius* and related species. Furthermore, based on the Aap gene cluster, the assembly machinery appears to be far simpler than that of its bacterial counterpart (Fig. 6). Whereas the bacterial T4P complex encodes for the alignment complex PilMNOP (Fig. 6b) and a secretin (PilQ) in diderm bacteria, the crenarchaeal Aap pilus lacks these components (Fig. 6a). It is plausible that this alignment complex is obsolete, as *S. acidocaldarius* does not have a second membrane. Instead, the Aap must traverse the S-layer—a porous, proteinaceous cage that surrounds many archaea. Whether a so-far elusive protein complex is required to guide the Aap through the S-layer or whether the S-layer itself acts as a guiding scaffold remains to be elucidated.

## Methods
### Cell culture and Aap isolation
Pilus isolation was performed as described in Gaines et al.[3]. Briefly, *S. acidocaldarius* strains (Table 1) were inoculated from cryo stock into $6 \times 5$ ml basal Brock at pH3, supplemented with 0.1% NZ-amine, 0.2% dextrin and 10 μg/ml Uracil. Cell cultures were grown for 48 h at 75 °C with light agitation. 10 ml of the resulting preculture were used to inoculate two litres of main culture and cells were grown to $OD_{600nm} = 0.5–0.8$. Cells were harvested by centrifugation at $5000 \times g$ for 25 min and 4 °C. Cell pellets were then resuspended in 20 ml Basal Brock medium (pH 3) without $FeCl_3$. Aap filaments were sheared as described in Henche et al.[33], or via a peristaltic pump (Gilson Minipuls), connected to a syringe needle with 1.10 mm in diameter and 40–50 mm in length (Braun GmbH). The samples were homogenised at 25 rpm for 1 h, before the syringe needle was swapped for narrower ones with 0.45 mm and 0.10 mm diameter for shearing at 25 rpm for 1 h. Sheared samples were then centrifuged at 12,000 × g for 25 min at 4 °C followed by a subsequent ultracentrifugation at 200,000 × g for 90 min and 4 °C. This pellet was then resuspended in 500 μl Basal Brock medium without $FeCl_3$ and layered on 4.5 ml CsCl (0.5 g/ml).

**Table 1 | Deletion of genes in S. *acid ocaldarius***

| Used strains | | |
|---|---|---|
| **Strain** | **Genotype** | **Source/Reference** |
| **MW158** | Deletion of *upsE* and *flaJ* in MW001 | Henche et al.[95] |
| **MW153** | Deletion of *aapA* in MW001 | Henche et al.[34] |
| **MW154** | Deletion of *aapB* in MW001 | Henche et al.[34] |
| **MW161** | Deletion of *aapA* and *aapB* in MW001 | Henche et al.[34] |
| **MW114** | Deletion of *pibD* in MW001 | Henche et al.[11] |

Density gradient centrifugation was carried out at $250,000 \times g$ for 16 h and 4 °C and the resulting white band in the upper third of the tube was collected. This band was diluted with 5 ml of Basal Brock without $FeCl_3$ and pelleted at $250,000 \times g$ for 1 h at 4 °C. The pellet was finally resuspended into 150 µl citrate buffer (25 mM sodium citrate/citric acid, 150 mM NaCl, pH 3) and stored at 4 °C.

### Negative stain transmission electron microscopy of *S. acidocaldarius* cells

A 5 µl suspension of *S. acidocaldarius* cells was applied to freshly glow-discharged 300 mesh carbon coated copper grids (Plano GmbH, Wetzlar Germany), followed by an incubation period of 30 s at room temperature. Excess liquid was blotted away and cells were re-applied on the grid. This was repeated three times. Grids were stained with 2% Uranyl acetate. A Hitachi HT8600 transmission electron microscope (TEM) operated at 100 kV and equipped with an EMSIS XAROSA CMOS camera was used for imaging.

### Cryo-EM sample preparation and data collection

A 3 µl suspension of isolated *S. acidocaldarius* filaments (a mixture of Aap and threads) was pipetted onto glow-discharged 300 mesh copper R2/2 Quantifoil grids. Using a Vitrobot Mark IV (ThermoFisher Scientific), the grids were blotted for 5s with 597 Whatman filter paper, and a blot force of −1, in an environment of 95% relative humidity and a temperature of 21 °C, and subsequently plunge-frozen into liquid ethane. The grids were screened in a 120 kV FEI Tecnai Spirit TEM (Thermo Fisher Scientific), equipped with a Gatan OneView CMOS detector).

For high resolution data collection, a FEI Titan Krios electron microscope (Thermo Fisher Scientific), operating at a voltage of 300 kV was used. Images were recorded using a Gatan K2 Summit direct electron detector, operated in counting mode. Data were recorded at a calibrated magnification of 134,048× (relating to a pixel size value of 1.047 Å) using the EPU software package (Thermo Fisher Scientific). Movies were recorded at a dose of $0.77$ e-/Å$^2$ s$^{-1}$ at 40 frames s$^{-1}$, 10 s exposure, with an accumulated total dose of 42.33 e-/Å$^2$ and a set defocus range of −1.0 to −2.5 µm, using 0.3 µm steps. A total of 6272 movies were collected. Cryo-EM statistics are shown in Table 2.

### Cryo-EM image processing

Motion correction and CTF estimation were performed in cryoSPARC[47]. The e2helixboxer program from EMAN2[78] was used to manually pick Aap filaments and generate initial 2D classes in Relion[79]. These classes were then imported into cryoSPARC, as templates for the Filament tracer program. A filament diameter of 80 Å and a separation distance value of 0.6 was used for automatic filaments tracing. Initially, only the best 2D classes were selected but the quality of the resulting unbiased 3D helix refinement remained poor, likely due to a limited range of projections. To overcome this, a larger number of 2D classes was selected, corresponding to a total of 947,729 particles. Non-biased 3D refinement without the application of helical parameters subsequently resulted in map with 3.74 Å resolution. Using this map, the helical parameters were determined in real space suggesting values of

**Table 2 | cryoEM and model building statistics**

| | AAP (EMD-18119) (PDB-8Q30) |
|---|---|
| Data collection and processing | |
| Magnification | 105k |
| Voltage (kV) | 300 |
| Electron exposure (e–/Å$^2$) | 42.33 |
| Defocus range (µm) | −1.0 to −2.5 |
| Pixel size (Å) | 1.047 |
| Symmetry imposed | Twist: −39.859° Rise: 15.403 |
| Initial particle images (no.) | 2207,193 |
| Final particle images (no.) | 947,729 |
| Map resolution (Å) | 3.22 |
| FSC threshold | 0.143 |
| Map resolution range (Å) | 3.8-2.8 |
| Resolution (Å) (map/model; FSC = 0.5) | 3.22 |
| Refinement | |
| Initial model used (PDB code) | Ab initio |
| Model resolution (FSC = 0.50/0.143 Å) | |
| Model refinement resolution (Å) | 3.22 |
| Map sharpening B factor (Å$^2$) | 0 |
| Model composition | 72,072 |
| Non-hydrogen atoms | 10,152 |
| Protein residues | 408 |
| Glycans | |
| B factors (Å$^2$) | 64.8 |
| Protein | 205.6 |
| Glycan* | |
| R.m.s. deviations | 0.010 |
| Bond lengths (Å) | 1.611 |
| Bond angles (°) | |
| Validation | 1.43 |
| MolProbity score | 2.70 |
| Clashscore | 3.08 |
| Poor rotamers (%) | |
| Ramachandran plot | 99.0 |
| Favoured (%) | 1.0 |
| Allowed (%) | 0.0 |
| Disallowed (%) | |

*The refinement was conducted against the DeepEMhanced map, which blurred glycan densities, resulting in elevated B-factors for those.

15.4 Å rise and −39.9° twist. Local motion correction and CTF refinement were then carried out before a final helical refinement, this time applying the helical parameters. This resulted in a final map had a global resolution of 3.22 Å, estimated by Fourier shell correlation (FSC) between two independently refined half sets, using the Gold Standard correlation value of 0.143. The map was denoised and postprocessed using DeepEMhancer[80]. Local resolution was calculated within cryoSPARC[47], and maps were visualised in ChimeraX[81].

### Model building and validation

Initial manual model building in Coot enabled the unambiguous identification of AapB as the Aap subunit. This was possible due to the glycosylation pattern of AapB, which differs significantly from AapA. This assignment was corroborated by AlphaFold2[82] and ModelAngelo[49] predictions. MOLREP[83] was then used for phased molecular replacement to position the remaining monomers into the density. Changes in the rebuilt model were propagated using CCP4[84], so that the copied monomers fitted into their corresponding densities.

Coot was used to model the glycan structures[85], with an unusual sugars dictionary prepared in JLIGAND[86]. Refinement of the final structure was done using REFMAC5 via the CCPEM interface[87,88].

## Sequence analysis and structural prediction

The AapB sequence was loaded into SyntTax[62], in order to search for homologous sequences amongst other Thermoprotei species. The top six results were then compared against each other using Clustal Omega[89]. To visualise the predicted gene cluster surrounding AapB, the KEGG genome database was used[90], followed by SyntTax to search for homologous proteins in differing archaeal species[62]. Protein structure prediction was performed with AlphaFold2, using the online ColabFold tool[91].

## 3D variability analysis and molecular flexibility

The flexibility of the Aap and threads was analysed using the molecular 3D variability tools in cryoSPARC[47]. For both filaments, 20 frame flexibility modes were calculated and visualised in UCSF Chimera[92]. The model showing the most significant flexibility was used to build atomic models. In brief, the structure of the Aap was positioned into frame 0 of the map series using MOLREP and then refined against the maps from frame 1, 5, 10, and 20 using Phenix[93]. The generated models were then used to create a morph in ChimeraX[81].

## Reporting summary

Further information on research design is available in the Nature Portfolio Reporting Summary linked to this article.

## Data availability

The cryoEM map generated in this study has been deposited in the EM DataResource under accession code EMD-18119. The corresponding atomic coordinates have been deposited in the protein Data Bank database under accession code PDB 8Q30. The raw image data used in this study have been deposited to the Electron Microscopy Public Image Archive (EMPIAR) under accession number EMPIAR-11889. The *S. acidocaldarius* (DSM639) genome can be accessed via the KEGG accession code T00251 or the NCBI Genbank code CP000077 112. The transcriptomics data analysed in this study can be accessed via the NCBI Gene Expression Omnibus under accession code GSE18630 or in the Pan Genomic Database for Genomic Elements Toxic To Bacteria (https://exploration.weizmann.ac.il/TCOL/index_singleOrg.php?organism=sulfolobus_acidocaldarius&tab=0).

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

## Acknowledgements

We would like to thank Diamond Light Source for providing access to the cryoEM facilities at the UK national electron bio-imaging centre (eBIC), funded by the Wellcome Trust, MRC and BBSRC. eBIC access was granted under the BAG allocation EM18258. Filament preparations were initially screened at the EM facility of the Faculty of Biology at the University of Freiburg. The TEM (Hitachi HT7800) was funded by the DFG grant (Project-ID 426849454) and is operated by the University of Freiburg, Faculty of Biology, as a partner unit within the Microscopy and Image Analysis Platform (MIAP) and the Life Imaging Center (LIC), Freiburg. We also thank Patrick Tripp who established the filament isolation protocol. We also thank Ufuk Borucu for grid screening, and the GW4 Facility for High-Resolution Electron Cryo-Microscopy, funded by the Wellcome Trust (202904/Z/16/Z and 206181/Z/17/Z) and BBSRC (BB/R000484/1). BD, MG, MM and RUH were supported by an ERC Starting Grant under the European Union's Horizon 2020 research and innovation program (grant agreement No 803894), awarded to BD. MM was also funded by a BBSRC New Investigator Research Grant (BB/R008639/1) to VG. SS and SVA were supported by the Collaborative Research Centre SFB1381 funded by the Deutsche Forschungsgemeinschaft (DFG, German Research Foundation), Project-ID 403222702—SFB 1381. SS and SVA were also funded by the Deutsche Forschungsgemeinschaft (DFG, German Research Foundation) under Germany's Excellence Strategy (CIBSS – EXC-2189 – Project ID 390939984). CH was supported by the Agence Nationale de la Recherche (grants #ANR-16-CE16-0009-01 and #ANR-21-CE16-0021-01). For the purpose of open access, the author has applied a CC BY public copyright licence to any Author Accepted Manuscript version arising from this submission.

## Author contributions

Major contributions to (i) the concept or design of the study (S.A., B.D.,) (ii) the acquisition, analysis, or interpretation of the data (M.G., M.I., S.S., R.U.H., M.M., C.H., B.D.); (iii) writing of the manuscript (M.G., M.I., S.S., R.H., S.A., B.D.) and provision of resources (B.D., V.G., S.A.).

## Competing interests

The authors declare no competing interests.
