## [Peer Review File · Nature Communications]

CryoEM reveals the structure of an archaeal pilus involved in twitching motility.Reviewer #1 (Remarks to the Author):

In the manuscript from Gaines et al., the authors describe the cryoEM structure of the archaeal Aap pilus involved in twitching motility from *Sulfolobus acidocaldarius*. The pili were sheared off from an *S. acidocaldarius* mutant strain lacking archaella and UV inducible pili, subsequently purified via centrifugation and vitrified on EM grids. The raw cryoEM micrographs revealed homogenous filaments with a high degree of flexibility.

By first using an unbiased refinement strategy, the authors could define the helical parameters of this filamentous structure to -39° helical twist with a rise of 15.4 Å. Applying these parameters in a helical reconstruction, they further solved the structure of the Aap pilus at 3.2 Å resolution. Importantly, these parameters differ from previously solved homologous structures of adhesive type 4 pili. By an ab initio atomic model-building approach, which was guided by large side chains as well as glycosylation sites, it became evident that the pilus is solely composed of the protein AapB, which adopts three different conformations within the solved structure.

While the here reported structure of the Aap pilus revealed an unexpected domain organization, the manuscript resides on a rather descriptive level and is only corroborated by additional computational analysis. Furthermore, the manuscript appears to be somewhat rushed, which becomes apparent in several sloppy typos, mislabeling, and unclear figure design.

Major comments:

- This manuscript was submitted back-to-back on bioRxiv with the manuscript of Liu et al.. While both groups present similar structures, the main difference is the orientation of the globular C-terminal domain with respect to the N-terminal helix. The authors should show the density of the linker region together with a comprehensive analysis to exclude an error in their model. Without the cryoEM map available, this analysis is not possible for the reviewer.

- Line 140: The term "erroneous helical parameters" is misleading or at least has to be specified. While these parameters are erroneous for the here described structure, they don't have to be wrong in the already published structures.

-Line 154 and Figure S7: It is not clear to me, if this data was created during the course of this study or re-evaluated from Henche et al.? What strain was used here? An archellum deficient strain? Please specify these details more clearly in the figure legend. How can you be sure that the filaments seen in the negative stain EM images are Aap and not threads?

- Line 165-167: Using an AlphaFold2 prediction to confirm the ab initio model might not be the best choice. If the authors wish to use it, they should at least include the AlphaFold predictions of both AapA and AapB for comparison. Maybe an alternative computational approach such as ModelAngelo would be better suited to further corroborate their protein identification? Mass Spectrometry analysis of their purified sample could also give further hints if AapA is involved as minor pilin?

- Line 168-170: The authors suggest that AapA acts as a minor pilin, which may play a role in capping or nucleating the Aap. While the latter is imaginable, how can AapA be a nucleating factor, when an aapA-KO strain still assembles pili (see line 154-155)? It would be interesting to see if the aapA-KO strain is still capable of twitching motility.

- Figure 5: The color code of the predicted/solved structures does not follow the color code of the model and like this, it is hard to follow and understand. A uniform color code would facilitate the readability of this figure.

Minor comments:

- In general, many figures would benefit from labels, which would tremendously facilitate their analysis. Just to mention a few: add used strains in Fig. S7, add name of the pili for structural comparison in Fig. 3, add label for pilus type in Fig. 4a, labels on top of the models in Fig 5, etc.

- Figure 1, figure legend panel c: "cross section of a". I guess it is a cross section of b?

-Fig S3, figure legend panel f: "showed that the archaellum was composed...". I guess the authors mean Aap?

- Line 183: I guess here is a typo and it should be: the RMSD between A and C instead S and B?

-Line 232: Typo: "the map of the Aap pilus contained only contained".

- Glycosylation paragraph: Are there any O-glycans?

- Figure 4e,f: It is hard to read what is written in green letters on top of the 2D classes. What are these numbers?

- Line 258: Typo: c-terminal instead of C-terminal

- Line 286: Typo: Despite this, the all identified homologs... delete "the"

- The bioRxiv manuscript from Liu et al. should be cited.

Reviewer #2 (Remarks to the Author):

In this manuscript, Gaines et al presented their work in the interesting field of archaeal flagellum and pilus. Their work mainly focused on solving the structure of *Sulfolobus acidocaldarius* archaeal adhesive pilus (Aap) by helical cryo-EM and is accompanied by molecular dynamic simulations and AI modeling. While many pili structures are helical assemblies made from a single component in a single conformation, the authors show here that the one component, AapB, that comprises *S. acidocaldarius* Aap adopts three conformations instead. These conformations are well distinguished and resolved by using a reduced helical symmetry during the refinement.

While these findings are potentially interesting to the general audience of Nature Communications, and the beautiful features shown in the figures for the map (e.g., Fig 1a) match the expected landmarks of a 3.0 Å-ish resolution cryo-EM map, the manuscript certainly needs some tune-up before it could be accepted for publication in Nature Communications.

There are details that need to be addressed to make this manuscript solid. For example, an archaeal flagellum is not to be confused with a type IV pilus. Archaeal flagella were shown to be organized in a 10-strand helix vs type IV pili in a 7-strand helix. Furthermore, some technical ideas for cryo-EM need to be clarified and the corresponding results fortified by additional illustrations:

Most importantly, a helical structure can not be reliably determined in an "unbiased" way, as described in the manuscript. Briefly, since the segments of helical filaments share the same common-line in Fourier space, unless the filaments are highly tilted in ice, determination of relative orientation for them is risky when solely done by a projection matching refinement method. Although the Cryosparc tutorial, for example, shows the "unbiased" refinement of MAVS/CARD, the very same structure was once reported in a wrongly indexed helix.

Therefore, if the authors really want to use this "unbiased" reconstruction method, at least they have to show some validation of the determined structure / determined helicity, e.g., by comparing projections of the map with 2D classification of the segments. They should also show the good matching between densities and models in the areas seemingly related by the quasi 3-fold symmetry but indeed nonequivalent (for example, the β -sheet region above S37 for all three conformations). In fact, Fourier space indexing is the unambiguous method to estimate any helical symmetry for such a structure.

I am open to see a revised manuscript after these points and the following minor points have been

addressed by the authors.

Minor points:

1. Results, lines 129-142, the authors should illustrate, perhaps in words, the relationship between 3 times 105 and -39, which is not readily clear to the general audience.
 2. Same, line 139, "the FFT was scrambled". Do you mean that the resolution was reduced or the features (reflections/layer-lines) were not as good after the operation?
 3. Line 145, the authors should state the average resolution shown by the FSC curves (Suppl. Fig. 5a).
 4. Line 186, the term "melted region" may not be understood readily by the general audience.
 5. Line 245, the protein assembly "threads" should be addressed in the introduction.
 6. Line 305 and below, section "A model of the Aap Machinery", the author may use more space to address how the AI generated models are related to the functions of the machinery, or refer the reader to the relevant paragraphs in Discussion.
 7. Discussion, line 365, "three pilins", I suppose the authors mean "three conformers of (the same) pilins".
 8. Methods, line 454, "Thermo Fisher Scientific" is not registered in the Netherlands.
 9. Methods, line 464 and below, section "Cryo-EM image processing", this reviewer finds that the details in this method section are insufficient and hard to follow. For example, the inter-box distance between two consecutive segments is crucial for relating the number of particles to number of unique asymmetric units.
 10. Fig. 3 (d-f), the panels should make it easy to see that the left two assemblies (d,e) are 7-strand helices and the right assembly (f) is a 10-strand helix.
- Suppl. Fig. 12 (f-j), are these newly determined structures or previously published work? If new, please deposit the map and model and list them in the "Data Availability" section. If published, please cite the reference and PDB/EMDB code in the legends.

Reviewer #3 (Remarks to the Author):

The manuscript by Gaines et al. presented structures of the archaeal pilus. The authors described that the pilus protein, AapB, has three different conformations when forming the pilus, and these contribute to the stability required during twitching by pilus. This study advanced our understanding of the mechanism of pilus. A few concerns should be addressed before being considered for publication.

Major:

1. In this paper, the differences in the three conformations are a particularly intriguing feature that is not found in other Type IV pili. Specifically, the fact that these 3-start helices exhibit consistent structural features, as opposed to the 11-start helices (along the protofilament) observed in Salmonella's hook and filament, suggests the involvement of distinctive interactions in a unique formation mechanism. The description mentions that the differences in the three conformations are attributed to the partial dissolution of α -helix, but it is important to provide an explanation for why such differences occur. For instance, it should be addressed whether there are variations in interactions with adjacent subunits or other factors contributing to these distinctions.
2. In the mention of glycosylation started from P6 I212, it would be extremely difficult to identify the glycans from a map such as the one shown in Supplemental Figure 12 without any support of biochemical experiments. In particular, the density of glycan at N88 cannot be identified from Supplemental Figure 12d. Furthermore, the reader has no way to know this difference, as there is no figure showing the absence of the N114 glycan. The authors should describe how you assessed the presence or absence of glycans in your criterion.
3. Figures 4a-d do not provide a focused view on the glycans in certain regions, and therefore, it is

advisable to create more zoomed-in figures. Especially, more focus should be given to glycan resembling wedges that should be represented in the thread.

Minor:

1. P6 I183 – “conformations S and B...” <- It is probably “conformations B and C ...”
2. P6 I191 – “ β -strand (β 1) following residue G37 (Fig. 2c).” <- There is no “ β 1 character ” in Fig. 2. Authors should remove the “(β 1)” or refer to Supplementary Figure 10. And Fig. 2c should be modified to Figure 2c
3. P7. I232 – “sugar residue (Figure 12 a-e).” <- It is probably “sugar residue (Supplementary Figure 12 a-e).”
4. P8 I274 – “adjacent subunits (Figure 4 f, ...” <- It probably “adjacent subunits (Figure 4 d,”.
5. The reference 61 was already published. The authors should refer to the original paper in NSMB, 2019.
6. Since the outline of the arrowhead in Supplementary Figure 2 is too wide, the color is difficult to see.
7. In the legend of Supplementary Figure 2, “archaella (orange arrowheads; a and b) and threads (white arrowheadss; b) <- orange arrow heads are not in b. And There are too many s in the “white arrowheadss; b)”

Rebuttal

Dear reviewers,

Many thanks for your time and effort to review our manuscript entitled “**CryoEM reveals the structure of an archaeal pilus involved in twitching motility**”. We appreciate all of your comments, addressed each of them thoroughly, and provided a point-by-point response below.

We believe that our manuscript has much improved and hope that it is now acceptable for publication.

Yours sincerely,

Bertram Daum

Reviewer's comments in **black**, authors responses in **blue**.

Reviewer #1 (Remarks to the Author):

In the manuscript from Gaines et al., the authors describe the cryoEM structure of the archaeal Aap pilus involved in twitching motility from *Sulfolobus acidocaldarius*. The pili were sheared off from an *S. acidocaldarius* mutant strain lacking archaella and UV inducible pili, subsequently purified via centrifugation and vitrified on EM grids. The raw cryoEM micrographs revealed homogenous filaments with a high degree of flexibility.

By first using an unbiased refinement strategy, the authors could define the helical parameters of this filamentous structure to – 39° helical twist with a rise of 15.4 Å. Applying these parameters in a helical reconstruction, they further solved the structure of the Aap pilus at 3.2 Å resolution. Importantly, these parameters differ from previously solved homologous structures of adhesive type 4 pili. By an ab initio atomic model-building approach, which was guided by large side chains as well as glycosylation sites, it became evident that the pilus is solely composed of the protein AapB, which adopts three different conformations within the solved structure.

While the here reported structure of the Aap pilus revealed an unexpected domain organization, the manuscript resides on a rather descriptive level and is only corroborated by additional computational analysis. Furthermore, the manuscript appears to be somewhat rushed, which becomes apparent in several sloppy typos, mislabeling, and unclear figure design.

Major comments:

- This manuscript was submitted back-to-back on bioRxiv with the manuscript of Liu et al.. While both groups present similar structures, the main difference is the orientation of the globular C-terminal domain with respect to the N-terminal helix. The authors should show the density of the linker region together with a comprehensive analysis to exclude an error in their model. Without the cryoEM map available, this analysis is not possible for the reviewer.

We have updated the figure to show maps and models of the linker region of the three AapB conformations. Note that we have refined our model of the Aap in the meantime, so the structures of these hinge regions have slightly changed. Maps and models have also been uploaded to this resubmission and deposited in the PDB and EMDB under the accession codes PDB 8Q30, EMD-18119.

- Line 140: The term “erroneous helical parameters” is misleading or at least has to be specified. While these parameters are erroneous for the here described structure, they don’t have to be wrong in the already published structures.

We changed the wording in this section, making sure that it does not infer that previously published structures were the result of erroneous helical parameters.

-Line 154 and Figure S7: It is not clear to me, if this data was created during the course of this study or re-evaluated from Henche et al.? What strain was used here? An archellum deficient strain? Please specify these details more clearly in the figure legend. How can you be sure that the filaments seen in the negative stain EM images are Aap and not threads?

The data were created during this study. The $\Delta aapA$ knockout mutant is strain MW153, the $\Delta aapB$ knockout mutant is strain MW154 and the double knockout mutant $\Delta aapAB$ is MW161. We included this information in the figure legends. These strains are based on the “wild type” MW001, meaning that MW153 and MW154 still form archaella, threads and Ups. Archaella, Aap and threads have very different diameters (12 nm, 8 nm and 4 nm, respectively) and can thus be clearly distinguished by negative stain EM (see Supplementary figure 2).

- Line 165-167: Using an AlphaFold2 prediction to confirm the ab initio model might not be the best choice. If the authors wish to use it, they should at least include the AlphaFold predictions of both AapA and AapB for comparison. Maybe an alternative computational approach such as ModelAngelo would be better suited to further corroborate their protein identification? Mass Spectrometry analysis of their purified sample could also give further hints if AapA is involved as minor pilin?

We have included a comparison between AapA and AapB in Supplementary Figure 8, showing that AapA can be clearly distinguished from AapB. AapA is predicted to have longer β -strands 1 and 2, as well as a shorter interconnecting loop between these strands. In addition, AapA is predicted to be glycosylated at different sites than AapB. The new Supplementary Figure 9 shows that the sequence suggested by ModelAngelo has 77 % identity with AapB, but only 49% identity with AapA, further corroborating that AapB is the correct subunit candidate. Unfortunately, mass spectrometry cannot be performed, as Aap are resistant to digestion with trypsin or treatment with guanidine hydrochloride (Wang, et al. Nat Microbiol., 2019; Wang et al, Nat Commun. 2020). The same is true for the thread filament (Gaines et al, Nat Commun., 2022).

- Line 168-170: The authors suggest that AapA acts as a minor pilin, which may play a role in capping or nucleating the Aap. While the latter is imaginable, how can AapA be a nucleating factor, when an aapA-KO strain still assembles pili (see line 154-155)? It would be interesting to see if the aapA-KO strain is still capable of twitching motility.

We agree with the reviewer that it is unlikely that AapA acts as an (essential) nucleating subunit, as the knockout still assembles Aap. We have rephrased the sentence accordingly. Indeed, the back-to-back submitted paper by Charles-Orszag et al (BioRxiv Doi: 10.1101/2023.08.04.552066) demonstrates that AapA-deficient mutants show reduced twitching motility. We have now clearly highlighted this in our manuscript.

- The color code of the predicted/solved structures does not follow the color code of the model and like this, it is hard to follow and understand. A uniform color code would facilitate t

Figure 5 has been edited with a consistent colour code.

Minor comments:

- In general, many figures would benefit from labels, which would tremendously facilitate their analysis. Just to mention a few: add used strains in Fig. S7, add name of the pili for structural comparison in Fig. 3, add label for pilus type in Fig. 4a, labels on top of the models in Fig 5, etc.

We thank the reviewer for their suggestion and have labelled all figures more clearly. We have also attempted to redesign key figures to make them easier to interpret.

- Figure 1, figure legend panel c: “cross section of a”. I guess it is a cross section of b?

Correct. We have amended the legend of the redesigned Figure 1 accordingly.

-Fig S3, figure legend panel f: “showed that the archaellum was composed...”. I guess the authors mean Aap?

Thank you for highlighting this error –we corrected it.

- Line 183: I guess here is a typo and it should be: the RMSD between A and C instead S and B?

Thank you for spotting the typo. We have corrected it.

-Line 232: Typo: "the map of the Aap pilus contained only contained".

Thank you for spotting the typo. We have corrected it.

- Glycosylation paragraph: Are there any O-glycans?

O-glycosylation is not known to exist in *Sulfolobus acidocaldarius*. We also do not see any evidence for large modifications of serine or threonine residues. We have included a clarifying sentence in the manuscript.

- Figure 4e,f: It is hard to read what is written in green letters on top of the 2D classes. What are these numbers?

The green values at the top of the boxes showed the number of particles in each class, and the bottom values showed a resolution estimate in Ångström. However, we realise that these numbers do not significantly add to the message of the figure and have therefore removed them.

- Line 258: Typo: c-terminal instead of C-terminal

Thank you for spotting the typo. We have corrected it.

- Line 286: Typo: Despite this, the all identified homologs... delete "the"

Thank you for spotting the typo. We have corrected it.

- The bioRxiv manuscript from Liu et al. should be cited.

Agreed. We have now cited the manuscript.

Reviewer #2 (Remarks to the Author):

In this manuscript, Gaines et al presented their work in the interesting field of archaeal flagellum and pilus. Their work mainly focused on solving the structure of *Sulfolobus acidocaldarius* archaeal adhesive pilus (Aap) by helical cryo-EM and is accompanied by molecular dynamic simulations and AI modeling. While many pili structures are helical assemblies made from a single component in a single conformation, the authors show here that the one component, AapB, that comprises *S. acidocaldarius* Aap adopts three conformations instead. These conformations are well distinguished and resolved by using a reduced helical symmetry during the refinement.

While these findings are potentially interesting to the general audience of Nature Communications, and the beautiful features shown in the figures for the map (e.g., Fig 1a) match the expected landmarks of a 3.0 Å-ish resolution cryo-EM map, the manuscript certainly needs some tune-up before it could be accepted for publication in Nature Communications.

There are details that need to be addressed to make this manuscript solid. For example, an archaeal flagellum is not to be confused with a type IV pilus. Archaeal flagella were shown to be organized in a 10-strand helix vs type IV pili in a 7-strand helix.

We agree with the reviewer that archaeella and type IV pili are not the same thing. However, there is consensus in the field that archaeella and T4P share a common origin and belong to the same superfamily (e.g.). We have now made this clearer in our manuscript.

Furthermore, some technical ideas for cryo-EM need to be clarified and the corresponding results fortified by additional illustrations:

Most importantly, a helical structure can not be reliably determined in an “unbiased” way, as described in the manuscript. Briefly, since the segments of helical filaments share the same common-line in Fourier space, unless the filaments are highly tilted in ice, determination of relative orientation for them is risky when solely done by a projection matching refinement method. Although the Cryosparc tutorial, for example, shows the “unbiased” refinement of MAVS/CARD, the very same structure was once reported in a wrongly indexed helix.

Therefore, if the authors really want to use this “unbiased” reconstruction method, at least

they have to show some validation of the determined structure / determined helicity, e.g., by comparing projections of the map with 2D classification of the segments. They should also show the good matching between densities and models in the areas seemingly related by the quasi 3-fold symmetry but indeed nonequivalent (for example, the β -sheet region above S37 for all three conformations). In fact, Fourier space indexing is the unambiguous method to estimate any helical symmetry for such a structure.

We are well aware that it is essential to exercise great caution when performing helical reconstruction, whether it is done via Forier Bessel indexing or via an “unbiased” approach such as that available through cryoSPARC. Indeed, we believe that combining both approaches minimises the risk of obtaining erroneous reconstructions. We have recently published a showcase whereby relaxing the symmetry during helical reconstruction can reveal complex symmetries otherwise overlooked by layer line indexing (Gambelli, Isupov, Daum, Faraday Discussions, 2022). In addition, variations of this unbiased approach can provide new insights into curved or superhelical filaments, as exemplified by Kreutzenberger et al, Science, 2022.

We took great care to validate our approach by (i) analysing and comparing the layer line profiles of 2D classes with reconstructions, (ii) helical symmetry search, as well as (iii) scrutinising the quality of the resulting maps. As requested by the reviewer, we have included a new figure, showing the similarity between the 2D classes and the final reconstruction of the map (Supplementary Figure 4 a,b). We also included closeups of the map of the filament with the three conformations of AapB highlighted (Supplementary Figure 4 c) and multiple views of each pilin conformation (Supplementary Figure 4 d), demonstrating the quality of the non-equivalent maps in areas that are related by the quasi-3-fold symmetry. Supplementary Figure 4 e shows the map/model fit in these areas and Figure 2C shows closeups of maps and models of the loop region.

I am open to see a revised manuscript after these points and the following minor points have been addressed by the authors.

Minor points:

1.Results, lines 129-142, the authors should illustrate, perhaps in words, the relationship between 3 times 105 and -39, which is not readily clear to the general audience.

We clarified this in the results section.

2. Same, line 139, “the FFT was scrambled”. Do you mean that the resolution was reduced, or the features (reflections/layer-lines) were not as good after the operation?

We acknowledge that “scrambled” is slightly ambiguous and clarified this now in the results section. In fact, the layer line pattern changes upon applying “wrong” helical parameters.

3. Line 145, the authors should state the average resolution shown by the FSC curves (Suppl. Fig. 5a).

Thank you for flagging this omission. An FSC has been added to Supplementary Fig. 5.

4. Line 186, the term “melted region” may not be understood readily by the general audience.

Acknowledged. We reworded the sentence and refer to this region as “loop” or “hinge”.

5. Line 245, the protein assembly “threads” should be addressed in the introduction.

We attempted to clarify this more in the introduction.

6. Line 305 and below, section “A model of the Aap Machinery”, the author may use more space to address how the AI generated models are related to the functions of the machinery, or refer the reader to the relevant paragraphs in Discussion.

We attempted to present our model without over-interpreting it, so did not go into too much functional speculation pertaining to the AlphaFold models.

7. Discussion, line 365, “three pilins”, I suppose the authors mean “three conformers of (the same) pilins”.

Thank you for spotting this error – we now corrected this.

8. Methods, line 454, “Thermo Fisher Scientific” is not registered in the Netherlands.

Good point. We removed the country designation.

9. Methods, line 464 and below, section “Cryo-EM image processing”, this reviewer finds that

the details in this method section are insufficient and hard to follow. For example, the inter-box distance between two consecutive segments is crucial for relating the number of particles to number of unique asymmetric units.

We added this information in the Methods section and attempted to reword it so that it is easier to follow.

10. Fig. 3 (d-f), the panels should make it easy to see that the left two assemblies (d,e) are 7-strand helices and the right assembly (f) is a 10-strand helix.

Done. While for the *S. islandicus* pilus and the *M. villosus* archaeellum, a clear 7- or 10-fold strands can be seen in end on view, this is not the case for the *S. acidocaldarius* Aap.

Suppl. Fig. 12 (f-j), are these newly determined structures or previously published work? If new, please deposit the map and model and list them in the "Data Availability" section. If published, please cite the reference and PDB/EMDB code in the legends.

These are views of our previously published structure of the *S. acidocaldarius* thread. We have now clarified this in the legend of the re-designed figure.

Reviewer #3 (Remarks to the Author):

The manuscript by Gaines et al. presented structures of the archaeal pilus. The authors described that the pilus protein, AapB, has three different conformations when forming the pilus, and these contribute to the stability required during twitching by pilus. This study advanced our understanding of the mechanism of pillus. A few concerns should be addressed before being considered for publication.

Major:

1. In this paper, the differences in the three conformations are a particularly intriguing feature that is not found in other Type IV pili. Specifically, the fact that these 3-start helices exhibit consistent structural features, as opposed to the 11-start helices (along the protofilament) observed in *Salmonella*'s hook and filament, suggests the involvement of distinctive interactions in a unique formation mechanism. The description mentions that the differences in the three conformations are attributed to the partial dissolution of α -helix, but it is important to provide an explanation for why such differences occur. For instance, it should be addressed whether there are variations in interactions with adjacent subunits or other

factors contributing to these distinctions.

Thank you for raising this intriguing point, which we have addressed in the new Figure 3. In essence, the differences in the linker region result in different lengths of the α -helical tails. Note that we refined the Aap model during the revision period and now report on slightly different structures of the hinge regions. Regardless, the key messages remain the same. Conformation C, which has the longest loop, reaches further into the core of the filament. Consequently, the N-terminal residues (L16) belonging AapB subunits of the conformation C line up along a central axis of the filament. The L16 residues of conformations A and B do not reach as far into the centre, meaning that their L16 residues appear to spiral around the central axis created by conformation C.

2. In the mention of glycosylation started from P6 I212, it would be extremely difficult to identify the glycans from a map such as the one shown in Supplemental Figure 12 without any support of biochemical experiments. In particular, the density of glycan at N88 cannot be identified from Supplemental Figure 12d. Furthermore, the reader has no way to know this difference, as there is no figure showing the absence of the N114 glycan. The authors should describe how you assessed the presence or absence of glycans in your criterion.

We clarified the reviewer's questions in the revised manuscript and prepared new figures to make the glycan aspect clearer to the reader. The new figures are Figure 1 (particularly c, e and g), Figure 2 (particularly b and d) and Supplementary Figure 12. In brief, *S. acidocaldarius* generates one type of N-linked glycan and its sequence has been determined by mass spectrometry (Zähringer et al, Eur J Biochem, 2000; Peyfoon et al, Archaea, 2010; Meyer, et al, Mol Microbiol 2011). We have previously built the structure of this glycan into the cryoEM map of the *S. acidocaldarius* thread (Gaines et al, 2022), where glycan densities were particularly well resolved (see re-designed Supplementary Figure 12 e-i). While in the Aap the glycans appear to be more flexible, we could still resolve up to three sugar moieties linked to three surface glycans (N63, N75 and N103; Supplementary Figure 12 a-c), and we modelled only those for which we could see density. Note that the numbering of the residues has changed, as in our newly refined Aap model, we now count the N-terminal residue starting with the number 16, instead of 1 (the first 15 residues are post-translationally cleaved). In contrast to the above-mentioned residues, N129 does not show any signs of a glycan density (Supplementary Figure 12d).

3. Figures 4a-d do not provide a focused view on the glycans in certain regions, and

therefore, it is advisable to create more zoomed-in figures. Especially, more focus should be given to glycan resembling wedges that should be represented in the thread.

Done. We have provided zoomed-in views.

Minor:

1. P6 I183 – “conformations S and B...” <- It is probably “conformations B and C ...”

Thank you for spotting this typo – we have corrected it.

2. P6 I191 – “ β -strand (β 1) following residue G37 (Fig. 2c).” <- There is no “ β 1 character ” in Fig. 2. Authors should remove the “(β 1)” or refer to Supplementary Figure 10. And Fig. 2c should be modified to Figure 2c

Thank you, we have addressed this in the revised manuscript.

3. P7. I232 – “sugar residue (Figure 12 a-e).” <- It is probably “sugar residue (Supplementary Figure 12 a-e).”

Thank you for spotting this typo – we have corrected it.

4. P8 I274 –“adjacent subunits (Figure 4 f, ...” <- It probably “adjacent subunits (Figure 4 d,”.

Thank you for spotting this typo – we have corrected it.

5. The reference 61 was already published. The authors should refer to the original paper in NSMB, 2019.

We have updated this reference.

6. Since the outline of the arrowhead in Supplementary Figure 2 is too wide, the color is difficult to see.

We have re-vamped the figure, which should now be clearer.

7. In the legend of Supplementary Figure 2, “archaella (orange arrowheads; a and b) and threads (white arrowheads; b) <- orange arrow heads are not in b. And There are too many s in the “white arrowheads; b)”

Thank you flagging this, we have sorted it.

Reviewer #1 (Remarks to the Author):

The authors have made an effort to improve the manuscript and have addressed my concerns in the revised version.

Some minor comments:

- Figure 4: Why do the authors show in Fig. 4b the "naked" core of Aap in *S. islandicus* with the N-terminal alpha-helices, while they choose a different representation in a and c? Could this be standardized?

- Typos:

Line 232: ... into the centre of the filament's centre

Line 273: During the building the atomic model of the Aap...

Line 361: ... that the two of the species....

Reviewer #2 (Remarks to the Author):

Previously Gaines et al presented a manuscript on the archaeal adhesive pilus (Aap) from *Sulfolobus acidocaldarius*. The authors now revised the manuscript according to the comments and suggestions from all three reviewers.

The revised manuscript addressed my main concern which is the technical validity of the method used in this study. The authors now showed an additional Supplementary figure (Suppl. Fig. 4) to illustrate the agreement between the representative 2D class averages from raw particles and representative projections of the final map. At the same time, panels in Suppl. Fig. 3 show good matching between the layer line image of the map from "unbiased refinement" and that from the final helical refinement. Other minor comments from my review have also been adequately addressed.

I am convinced that the manuscript is now ready for publication.

Reviewer #3 (Remarks to the Author):

The authors responded to reviewers' comments in good faith and made corrections and additions to the figures and manuscript as expected. I do not see the need for any further amendments.

REVIEWERS' COMMENTS

Reviewer #1 (Remarks to the Author):

The authors have made an effort to improve the manuscript and have addressed my concerns in the revised version.

We thank the reviewer for their positive evaluation and addressed their remaining minor comments below.

Some minor comments:

- Figure 4: Why do the authors show in Fig. 4b the “naked” core of Aap in *S. islandicus* with the N-terminal alpha-helices, while they choose a different representation in a and c? Could this be standardized?

For this filament structure (Fig. 4B, *S. islandicus* LAL14 AAP filament) only a short section of the filament has been modelled and deposited by Wang et al (2019) in the PDB databank (PDB-6NAV; www.rcsb.org/structure/6NAV). Hence, we are unable to show a longer section of the model to standardize the view with a and c.

- Typos:

Line 232: ... into the centre of the filament's centre

Fixed

Line 273: During the building the atomic model of the Aap...

Fixed

Line 361: ... that the two of the species....

Fixed

Reviewer #2 (Remarks to the Author):

Previously Gaines et al presented a manuscript on the archaeal adhesive pilus (Aap) from *Sulfolobus acidocaldarius*. The authors now revised the manuscript according to the comments and suggestions from all three reviewers.

The revised manuscript addressed my main concern which is the technical validity of the method used in this study. The authors now showed an additional Supplementary figure (Suppl. Fig. 4) to illustrate the agreement between the representative 2D class averages from raw particles and representative projections of the final map. At the same time, panels in Suppl. Fig. 3 show good matching between the layer line image of the map from “unbiased refinement” and that from the final helical refinement. Other minor comments from my review have also been adequately addressed.

I am convinced that the manuscript is now ready for publication.

We thank the reviewer for their positive evaluation.

Reviewer #3 (Remarks to the Author):

The authors responded to reviewers' comments in good faith and made corrections and additions to the figures and manuscript as expected. I do not see the need for any further amendments.

We thank the reviewer for their positive evaluation.